# PMIP4 experiments using MIROC-ES2L Earth System Model

Rumi Ohgaito[1], Akitomo Yamamoto[1], Tomohiro Hajima[1], Ryouta O'ishi[2], Manabu Abe[1], Hiroaki Tatebe[1], Ayako Abe-Ouchi[2,3,1], Michio Kawamiya[1]

[1]Research Center for Environmental Modeling and Application, Japan Agency for Marine-Earth Science and Technology, 3173-25 Showamachi, Kanazawaku, Yokohama 236-0001, Japan
[2]Atmosphere and Ocean Research Institute, University of Tokyo, Kashiwa 2778568, Japan
[3]National Institute of Polar Research, Tachikawa 1908518, Japan

*Correspondence to*: Rumi Ohgaito (ohgaito@jamstec.go.jp)

**Abstract.** Following the protocol of the fourth phase of the Paleoclimate Modeling Intercomparison Project (PMIP4), we performed numerical experiments targeting distinctive past time periods using the Model for Interdisciplinary Research on Climate, Earth System version 2 for Long-term simulations (MIROC-ES2L), which is an Earth System Model. Setup and basic performance of the experiments are presented.

The Last Glacial Maximum was one of the most extreme climate states during the Quaternary and conducting numerical modeling experiments of this period has long been a challenge for the paleoclimate community. We conducted a Last Glacial Maximum experiment with a long spin-up of nearly 9,000 years. Globally, there was reasonable agreement between the anomalies relative to present day derived from model climatology and those derived from proxy data archives while some regional discrepancies remained.

By changing orbital and greenhouse gas forcings, we conducted experiments for two interglacial periods: 6,000 and 127,000 years before present. Model anomalies relative to present day were qualitatively consistent with variations in solar forcing. However, anomalies in the model were smaller than those derived from proxy data archives, suggesting that processes that play a role in past interglacial climates remain lacking in this state-of-the-art model.

We conducted transient simulations from 850 CE to 1850 CE and from 1850 CE to 2014 CE. Cooling in the model indicated clear response to huge volcanic eruptions, consistent with paleo-proxy data. The contrast between cooling during the Little Ice Age and warming during the 20[th] to 21[st] centuries was represented well at the multidecadal timescale.

## 1 Introduction

Using climate models to simulate past climate provides unique opportunities to evaluate models' projections of future climate. The Paleoclimate Modeling Intercomparison Project (PMIP) began in the early 1990s (Joussaume et al., 1999). Since then, the paleoclimate community has continued to expand their research to include more time periods and events. With the increase of computational power, models of higher complexity are used to make future projections (Kawamiya et al., 2020). Phase 3 of PMIP was endorsed by phase 5 of the Coupled Model Intercomparison Project (CMIP5; Braconnot et al., 2012), and PMIP is

now in its fourth phase (PMIP4; Kageyama et al., 2018). The proposed PMIP4 experiments cover a wide range of time periods, including the Last Glacial Maximum (LGM; 21,000 years before present), two interglacials (6,000 and 127,000 years before

present), the last millennium (LM), the mid-Pliocene, and many non-CMIP time periods.

The Quaternary is characterized by cyclic climate change with long glacials and short interglacials that have been recorded in various paleo-proxy records such as ice cores (Jouzel et al., 2007; Dome Fuji Ice Core Project Members, 2017), ocean sediment cores (Weldeab et al., 2007), loess records (Maher et al., 2010), and terrestrial fossils (Bartlein et al., 2011). The LGM refers to the period when global ice volume reached its maximum. It was also one of the coldest periods of the Quaternary.

Since the beginning of PMIP, attention has been drawn toward the LGM, which was one of the extreme periods in the glacial–interglacial cycles of the Quaternary (Joussaume and Taylor, 2000; Braconnot et al., 2007; Kageyama et al., 2020), and also the most recent period during which global coverage of the continental ice sheets was at its maximum and greenhouse gas (GHG) levels were at a minimum.

As LGM cooling relative to the preindustrial (PI) experiment over the tropics is at a comparable level to equilibrium climate

sensitivity (ECS), LGM modeling can provide useful information to constrain climate sensitivity for projections of future climate (Annan et al., 2006; Renoult et al., 2020). Intercomparison studies of proxy-based reconstructions of climate variables and model output continue to be conducted (Braconnot et al., 2007; Bartlein et al., 2011; Kageyama et al., 2020). They report good agreement between model output and proxy data for temperature and sea surface temperature (SST) anomalies over the low latitudes (Otto-Bliesner et al., 2009; Hargreaves et al., 2013); however, the tendency for models to underestimate cooling

over Greenland remains (Masson-Delmotte et al., 2006). Models have difficulty in reproducing the weakened Atlantic meridional overturning circulation (AMOC) of the LGM (Weber, 2007; Brady et al. 2013; Muglia and Schmittner, 2015; Marzocchi and Jansen, 2017), which might influence the underestimation of cooling. The dust deposition was several to tens of times higher at LGM (Lambert et al. 2008, Lamy et al. 2014, Dome Fuji Ice Core Project members 2017), but was difficult to reproduce by LGM experiments; to reproduce the dust abundance at LGM, we need to assume glaciogenic dust (Mahowald

et al. 2006, Ohgaito et al. 2018), or assuming an erodibility map (Albani et al. 2014). And an erodibility map was formally introduced in PMIP4 (Kageyama et al. 2017), in addition to the dust emission that is simulated in non-Paleo simulations. In Ohgaito 2018, they showed that sufficient dust loading affects the temperature around Antarctica.

The interglacial periods of 6,000 and 127,000 years before present were characterized by differences in solar radiation at the top of the atmosphere caused by orbital states that were different from those of the present day (Brierley et al., 2020; Otto-

Bliesner et al., 2020), resulting in seasonalities that were different from the PI period (1850 CE). As it was in the recent past and because various paleo-proxy records are available (Ritchie et al., 1985; Drake et al., 2011; Hely et al., 2014; Tierney et al. 2013, 2017), the interglacial period of 6,000 years before present was the only interglacial included in earlier phases of PMIP (Braconnot et al., 2007; Ohgaito and Abe-Ouchi, 2007, 2009; Ohgaito et al., 2013). Following efforts to collect paleo-proxy data (Otto-Bliesner et al., 2001; Lunt et al., 2013; Capron et al., 2014, 2017), it is now also possible to conduct the same

experiment for 127,000 years before present; experiments on this interglacial period have the advantage of strong seasonality in the Northern Hemisphere (NH). The insolation anomaly at 127,000 years before present was larger than that at 6,000 years

before present, and the stronger insolation at 127,000 years before present during boreal summer modulated the temperature and circulation for that time period (Lunt et al., 2013). The role of vegetation coupling has been discussed intensively as studies report that vegetation enhances warming in the NH (O'ishi and Abe-Ouchi, 2011). However, models have been unable to
reproduce the quantitative changes recorded in proxy data (McKay et al., 2011, Capron et al., 2014, Hoffman et al., 2017).

As the LM is the most recent period prior to the PI period, there are vast amounts and varieties of data available from exact times in proxy records (PAGES2k-PMIP3 group, 2015; Luterbacher et al., 2016; Gagen et al., 2016) and in the literature (Pfister and Brazdil, 2006; Xoplaki et al., 2016; Camenisch et al., 2016). In earlier numerical paleoclimate studies, simple
models were used to conduct transient experiments over periods of 1,000 years (Crowley, 2000; Goosse et al., 2005). With the increase of computational power, simulations using coupled Atmosphere–Ocean General Circulation Models (AOGCMs) and/or comprehensive Earth System Models (ESMs) became standard (Kawamiya et al., 2020). Coordination of LM experiments began under PMIP3 (Schmidt et al., 2011), and multiple AOGCMs and ESMs have been used to perform LM experiments. One of the important questions for LM experiments is whether climate variabilities stem from internal variability
or forced responses. Atwood et al. (2016) decomposed the forcing of the LM experiment and concluded that cooling during the Little Ice Age (LIA; 1450–1850 CE) was largely driven by volcanic eruptions. PAGES2k (2015) summarized reconstruction–model intercomparisons and reported that the agreement between model and proxy-based reconstructions is better in the high latitudes in the NH and worse in the Southern Hemisphere. Historical (HIST; 1850–2014 CE) and LM experiments are intrinsically different from the other PMIP4 time-slice experiments discussed in this paper. They are time-
varying experiments that follow the same method used in the historical experiment in CMIP6. Hence, the LM experiment is closely aligned with the scientific focus of other endorsed MIPs, such as comparison of climatic response to volcanic forcing (VolMIP; Zanchettin et al., 2016) and land use (LUMIP; Lawrence et al., 2016).

Using the Model for Interdisciplinary Research on Climate, Earth System version 2 for Long-term simulations (MIROC-ES2L), we performed numerical experiments targeting distinctive time periods. These simulations included the LGM, the 6ka
and 127ka, and the LM experiments. The MIROC-ES2L is an ESM that contains atmosphere, ocean, land, and ocean and land biogeochemical cycles (Hajima et al. 2020), which has been developed recently to contribute to CMIP6 and the United Nations Intergovernmental Panel on Climate Change Sixth Assessment Report.

The model is presented in Sect. 2 and the experimental setup and spin-up procedures are explained in Sect. 3. Basic climate states from the experiments are presented in Sect. 4, and conclusions and an outlook are discussed in Sect. 5.

## 2 Model

The MIROC-ES2L is an ESM developed for CMIP6 (Hajima et al., 2020) and its physical core comprises atmosphere, ocean, and land modules; variables are exchanged via a flux coupler (Fig. 1). The AOGCM components are the same as those in Tatebe et al. (2018). The physical ocean and land modules are coupled with the land ecosystem model VISIT-e (Ito and Inatomi,

2012) and the ocean biogeochemical model OECO2 with a nutrient–phytoplankton–zooplankton–detritus type representation of the ecosystem. The ecosystem modules can simulate global carbon and nitrogen cycles explicitly. As the carbon and nitrogen in the atmosphere are prescribed to predetermined values in each experimental setting in this study, carbon and nitrogen variables calculated in OECO2 and VISIT-e are not returned to the atmosphere. Distribution of plant functional types (PFTs) is prescribed because VISIT-e is not a dynamic vegetation model. Dynamics of aerosols are calculated by an online aerosol module, SPRINTARS (Takemura et al., 2000, 2005, 2009).

Horizontal resolution of the atmosphere is set to T42 spectral truncation. Vertical resolution is 40 levels up to 3 hPa. The ocean component has tripolar horizontal coordinates, with two poles in the NH that are located over land to avoid singularity over ocean grids. Horizontal resolution of the ocean is 1° longitude and varies from 0.5° latitude around the Equator to 1° latitude over the mid-latitudes. Vertical resolution of the ocean is 62 layers with a hybrid sigma-z coordinate.

Using this model, various types of CMIP6 experiments have been performed. These include all of the Diagnostic, Evaluation and Characterization of Klima (DECK) experiments, the historical experiment of CMIP6 (from 1850 CE to 2014 CE), and the endorsed MIP experiments. The ECS of this model version is 2.66 K by Gregory's method (Gregory et al. 2004). The identical model version was used for all the experiments in this study.

## 3. Experimental setup and spin-up procedures

### 3.1 Setup and spin-up of preindustrial (PI) control experiment

The PI control experiment is the reference experiment of all the paleoclimate experiments. It is identical to the piControl experiment in CMIP6 (Eyring et al., 2016) and the experimental configuration of PI in MIROC-ES2L is described in detail in Hajima et al. (2020). Levels of GHGs were set following the protocol of CMIP6: $CO_2$, $CH_4$, and $N_2O$ were set to 284.725 ppm, 808.25 ppb, and 273.02 ppb, respectively (Table 1). The PFTs in PI are inherited from MIROC-ESM (Watanabe et al. 2011), which was based on Ramankutty and Foley (1999) (Fig. 2(c)). A description of each PFT is given in the caption of Fig. 2(c). Topography is defined from GTOPO30 (Fig. 2(e)). The experiment was run for more than 9,000 model years during the course of model development and the final drift of the global mean surface air temperature was $-4.79 \times 10^{-5}$ °C yr$^{-1}$ for the final 500 years (Hajima et al. 2020). Model output from this period was submitted to CMIP6 and the climatology of this period is used for the analyses in this study.

### 3.2 Setup and spin-up of Last Glacial Maximum (LGM) experiment

We performed the LGM experiment following PMIP4 protocol (Kageyama et al., 2017). A long spin-up is essential because of the considerable differences between LGM and present-day conditions. Hence, before model development was finalized, we started spinning up using the physical core (AOGCM) of MIROC-ES2L (Tatebe et al., 2018). Spin-up started by reducing $CO_2$ (Bereiter et al., 2015), $CH_4$ (Loulergue et al., 2008), and $N_2O$ (Schilt et al., 2010) levels from PI to LGM values (Table

1). Global mean air temperature gradually reached quasi-equilibrium. After integration for 2,640 model years, the land–sea mask, ice sheets, altitude (from ICE-6G_C, as presented in Peltier et al. (2015) (Fig. 2(b, d, and f))), river courses, and Earth's orbit (Berger, 1978) (Fig. 3(a)) were changed from PI to LGM conditions step by step. The total spin-up time was 6,760 model years (Figs. 4 and 5). The LGM PFTs were created based on the PI PFTs with the ice sheet grids defined by ICE-6G_C, and nearby PFTs were diverted to non-ice sheet land (exposed continental shelves) expanded from PI. In this process, the biggest

technical difficulty occurred when the LGM land–sea mask and ice sheet albedo were introduced ((2) of Fig. 5). As mentioned in Appendix, the LGM spin-up failed at year 3305 and the simulation could not proceed because a grid reached the upper limit of the model's sea ice thickness on the north coast of North America. Figure A1 shows the time evolution of winter sea ice thickness along the north coast of North America during the spin-up. In the present-day condition, the Beaufort gyre prevails on the north coast of North America, with southward winds pushing sea ice toward land. In addition to this, the sea ice thickness

has increased too much due to the cold conditions. When we changed the elevation to the LGM condition, which was left at PI ((3) of Fig. 5), the circulation field changed due to the appearance of the high altitude Laurentide ice sheet, and the north coast of North America had neutral north-south winds on average, which would eliminate the sea ice build-up phenomenon and allowed the experiment to continue.

        As the development of MIROC-ES2L was finalized during LGM spin-up, conditions in the 6,760[th] model year of the spin-up

were used to initiate the LGM experiment in MIROC-ES2L. In this conversion procedure, the offline terrestrial module was spun up for 40,000 model years until quasi-stability was reached and the end state was used in the LGM experiment in MIROC-ES2L. This was followed by spinning up the main MIROC-ES2L experiment for a further 100 years. Ocean salinity (1 Practical Salinity Unit (PSU) was added globally) and an erodibility map (addressing dust emission under LGM conditions, as proposed by Albani et al. (2014, 2016)) were introduced. The erodibility map specifies low latitudes as deserts and mid- to high latitudes

as tundra (Fig. 2(d)). Land and ocean ecosystem models were spun up offline for 40,000 and 3,000 model years, respectively, on the basis of the physical conditions created by MIROC-ES2L. Land and ocean biogeochemical states at the end of the offline spin-up were used to initialize the LGM experiment in MIROC-ES2L. The LGM experiment was run for a further 1,800 years until it eventually reached quasi-equilibrium. Surface air temperature of the final 500 model years showed a trend of 0.0002 °C yr$^{-1}$. Model output from the final 100 years was submitted to PMIP4-CMIP6.


### 3.3 Setup and spin-up of the two interglacial experiments

        The 6ka and 127ka experiments were spun up following the protocol outlined in Otto-Bliesner et al. (2017). The specified GHGs and orbital parameters were as listed in Table 1. The main difference between these periods and the PI period was the change in insolation attributable to Earth's orbit, as shown in Fig. 3(b and c), where seasonality was amplified in the NH and

diminished in the Southern Hemisphere.

        Starting from PI, the 6ka experiment was integrated for 1,500 model years and the 127ka experiment was integrated for 1,550 model years (Fig. 4(b and c)). After the long spin-up, the final 100 years of the simulations were selected as the formal products

to be submitted to CMIP6 and for analyses in this study. The 127ka experiment is identical to the LIG experiment in O'ishi et al. (2020).


### 3.4 Setup and spin-up of the Last Millennium and historical experiments

We performed the LM experiment following the protocol of Jungclaus et al. (2017). The experiment was forced with time-varying total solar irradiance (Shapiro et al., 2011; Vieira et al., 2011; Wu et al., 2017), orbit (Berger, 1978), GHGs (Meinshousen et al., 2017), volcanic eruptions (Sigl, 2015; Tooney and Sigl, 2017), ozone, and land use change (Hurtte et al., 170 2016) (Table 1). The experiment followed the same procedure as that of the historical experiment in CMIP6 (Eyring et al., 2016). From PI, the model was run under constant forcing from 850 CE for 200 model years (Fig. 4 (d)). The end state of the spin-up was used to initialize the time-varying LM experiment, which was conducted from 850 CE to 1850 CE. We performed a HIST experiment following CMIP6 protocol (Eyring et al., 2016). The end state of the LM experiment was used to initialize HIST, which was run until 2014 CE.

## 4. Comparison of mean climate states derived from model output and paleoclimate proxy data archives

### 4.1 PI mean climate

Hajima et al. (2019) analyzed basic model performance for the present and indicated that MIROC-ES2L is a state-of-the-art ESM that is able to reproduce mean climatology reasonably well. Global annual mean air temperature at 2 m height is 14.99 °C and the peak value of the annual mean AMOC is 15.3 Sv, which falls within the range of reasonable estimates (Frajka-Williams 180 et al., 2019). The peak AMOC is defined as the peak value of the area from 15°–60°N and from 900–3300 m depth. Model SST presents a reasonable global distribution but has positive bias over the Southern Ocean, which leads to underestimation of the extent of Antarctic sea ice.

### 4.2 LGM mean climate

Relative to PI, the final 100 years of the LGM has a global mean surface air temperature anomaly of −4.4 °C (Figs. 5(a) and 6(a)) and a tropical air temperature anomaly of approximately −2 °C, consistent with values derived from paleo-proxy archives (MARGO project members, 2009; Bartlein et al., 2011). Borehole thermometry suggested a temperature anomaly of the LGM relative to PI over Eastern Antarctica of −7 to −10 °C (Stenni et al., 2010; Uemura et al., 2012). The temperature anomaly in the model is approximately −6.0 °C, suggesting that cooling in the model is weak. For central Greenland, borehole thermometry 190 suggested a temperature anomaly of −21 to −25 °C (Cuffey et al., 1995; Johnsen et al., 1995; Dahl-Jensen et al., 1998), whereas the model temperature anomaly is −11.1 °C. The large discrepancy between ice core data and model output could partly be attributed to issues related to the modeling of the thermohaline state of the ocean (McManus et al., 2004; Curry and Oppo,

2005). It is well known that numerical models have difficulty in reproducing the sluggish thermohaline circulation of the LGM (Otto-Bliesner et al., 2007; Muglia and Schmittner, 2015; Sherriff-Tadano et al., 2017) that is suggested in proxy data (Lynch-

Stieglitz et al., 2007; Hesse et al., 2011). In our experiment, the peak value of the annual mean AMOC of the LGM is 21.0 Sv (Figs. 5(b) and 7), which is higher than that of PI. To address this issue, we will continue the experiment to identify the components that contribute to global cooling and those that contribute to cooling over the polar regions. The sharp gradient shown by the contour lines around 32°N would be caused by a strong and deep westerly boundary current and associated strong upwelling (Brady et al., 2013), which can be seen in the previous LGM modelling studies having strong AMOC (Brady

et al. 2013, Muglia and Schmittner 2015, Sherriff-Tadano et al. 2018).

Figure 6(b) shows the net precipitation anomaly relative to PI. Total precipitation is 1063 mm yr$^{-1}$ for LGM and 1166 mm yr$^{-1}$ for PI. Consistent with Bartlein et al. (2011), model precipitation has a general tendency to be lower for the LGM than for the PI period because the lower SSTs and colder climate of the LGM result in a weaker hydrological cycle, which is also shown in the weakened zonal water vapor transport (Fig. S1(b)). Large reductions in precipitation relative to PI are found in

areas that were covered by ice sheets during the LGM but were no longer ice-covered at PI, i.e., the areas covered by the Laurentide and Fennoscandian ice sheets. These large anomalies would be associated with the higher altitude of the ice surface relative to the ground surface when the ice sheets have disappeared. In the northern North Atlantic Ocean, large anomalies are associated with the southward expansion of sea ice during the LGM.

Anomalies of zonal mean oceanic potential temperature and salinity are shown in Fig. S2(a and d). In the Southern Ocean,

proxy data (Adkins et al., 2002) suggested anomalies of −2 °C and +2.5 PSU at around 3600 m depth. The underestimation might be attributed to too little sea ice formation, which would be related to a warm bias of the Southern Ocean (Hajima et al., 2020). In contrast, in the North Atlantic Ocean, the anomaly of salinity agrees with the proxy data (Adkins et al., 2002), whereas the temperature anomaly (−1 to −2 °C) is underestimated (−4 to −5 °C; Adkins et al. (2002)). A temperature that is too warm could possibly be attributed to a high state of AMOC.

In Fig. 8, we compare the dust deposition fluxes with data archives (Kohfeld et al., 2013, Albani et al., 2014). The distribution is also shown in Fig. S3. The model shows general consistency with the data archives globally, with positive bias in the PI over Antarctica and Greenland, and values that are insufficiently high in the Gobi and Taklamakan regions. The LGM shows better representation of the proxy data than the PI, with reasonable fluxes over Antarctica. However, it underestimates the high values that are abundant in the East Asian region and the high dust fluxes in North America. The LGM dust fluxes are shown

in Fig. S3(c) as a ratio of the LGM dust fluxes to those of the PI. The ratio is generally well represented globally, but the ratio is underestimated in South America and in regions of the South Atlantic downstream of the wind. The reason for the underestimation, as mentioned above, is probably overestimation of South American dust emission in PI.

Figure 9 shows the export production anomaly of the oceanic ecosystem of the LGM relative to PI with paleo-proxy data (Kohfeld et al., 2013) superimposed on model output. As proxy data provide qualitative information rather than quantitative

assessments, comparisons between model output and proxy data can only be used to evaluate the accuracy of the general direction of the model anomaly. Positive model anomalies over the low–mid-latitudes are consistent with the proxy data.

Negative anomalies over the high latitudes can be understood to reflect sea ice expansion during the LGM. Sea ice expansion during the LGM is underestimated in the model (Crosta et al., 1998) and could result in negative anomalies around Antarctica that have smaller absolute values than those indicated by proxy data (Fig. 9).

An excessively weak positive anomaly around 40°–50°S in LGM could be the result of dust emission being too high over South America in the PI experiment (Fig. 8). Mean global terrestrial gross primary production in the LGM experiment is 65% of that in the PI experiment, which is consistent with the estimates of Prentice et al. (2011) obtained using a dynamic global vegetation model.

The dissolved inorganic carbon content of the ocean in LGM is 629 Pg C less than in PI. Lowering the atmospheric $CO_2$ to 190 ppm and strengthening of the overturning circulation lead to extraction of a large amount of carbon out of the ocean. Conversely, increased solubility owing to cooling and enhanced biological carbon export because of increases in nutrient and iron supply from the ocean interior and dust lead to accumulation of carbon within the ocean. The former effects mainly contribute to carbon redistribution, resulting in reduction in the carbon content of 239 Pg C in the upper ocean and 390 Pg C in the deep ocean (Table 2 and Fig. S4). The simulated glacial ocean is therefore unable to explain the glacial–interglacial drawdown of atmospheric $CO_2$, which is similar to previous modeling studies (Buchanan et al., 2016). It should be noted that the effects of burial–nutrient feedback and carbonate compensation on the oceanic carbon cycle are not considered in this simulation because MIROC-ES2L does not include a sediment module.

As the dissolved oxygen cycle is the mirror image of the biological carbon cycle, reconstructed oxygen change is useful to constrain the respired carbon accumulation. We compared modeled oxygen changes from PI to LGM with qualitative and quantitative proxy data (Jaccard et al., 2016; Durand et al., 2018; Schmiedl and Mackensen, 2006; Hoogakker et al., 2015, 2018; Gottschalk et al., 2016; Lu et al., 2016; Bunzel et al., 2017; Umling and Thunell, 2018). The combination of cooler SST and enhanced AMOC increases the oxygen content by approximately 10 mmol m$^{-3}$ in both the upper and the deep ocean (Table 2). The simulated increases in oxygen are in reasonable agreement with the proxy data for the upper ocean, but contrast the proxy data for the deep ocean, which show a decrease in oxygen of more than 30 mmol m$^{-3}$ (Fig. S4). The model–proxy disagreements of deep ocean oxygen change result in underestimation of the accumulated respired carbon.

The simulated deep-water-mass age is younger during the LGM than during the PI by approximately 60 years (Table 2), indicating an increase in ventilation due to enhanced AMOC. However, proxy data show an increase in water-mass age of more than 1,000 years, suggesting reduced ventilation and weaker AMOC (Burke and Robinson, 2012; Curry and Oppo, 2005). Enhanced ventilation supplies oxygen-rich surface water to the deep ocean and simultaneously releases carbon accumulated in the deep water to the atmosphere. Therefore, we attribute the model–proxy disagreements of deep ocean oxygen change and underestimation of respired carbon accumulation to overestimation of ventilation. Our results suggest that weaker AMOC is required for reproducing the respired carbon accumulation and deoxygenation in the glacial deep ocean.

## 4.3 Mean changes in interglacial experiments

Figures 10 and 11 show air temperature and precipitation anomalies of 6ka and 127ka relative to PI. Because of the marked changes in seasonality in these time periods, adjustment of the calendar was applied and average anomalies were calculated for June–August (JJA) and December–February (DJF). Air temperature anomalies are positive over continental interiors in the mid–high latitudes in JJA for both 6ka (Fig. 10(a)) and 127ka (Fig. 11(a)) because of changes in shortwave radiation forcing. Compared with 6ka, stronger shortwave forcing results in larger air temperature anomalies in 127ka (Fig. 3).

Precipitation anomalies suggest that, relative to PI, boreal summer monsoons were stronger (Figs. 10(b) and 11(b)) and austral summer monsoons were weaker during the interglacials (Figs. 10(d) and 11(d)). The precipitation anomaly over the Sahel region suggests that the reduction of desert area during the interglacials relative to PI is smaller in the model than that suggested by proxy data (Petit-Maire, 1999; Castaneda, 2009; Tierney et al., 2017; Drake et al., 2011; Hely et al., 2014), which is a mismatch that has been persistent through many modeling efforts (Braconnot et al., 2001, 2007). The zonal water vapor
transport is shown in Fig. S1(c and d). Relative to PI, more water vapor is transported to North Africa, and the amplitude is more pronounced in 127ka than 6ka following the radiation anomaly.

   Variations of temperature and precipitation anomalies with season and latitude are shown in four Hovmöller diagrams in Fig. 12. The temperature anomaly responds to changes in solar radiation with a lag of approximately one month (Figs. 3(b and c) and 12(a and b)), which could be a consequence of the slow thermal response of the ocean surface. Precipitation anomalies of
6ka and 127ka, relative to PI, exhibit a northward shift and enhancement during boreal summer in the NH (Fig. 12(c and d)) which is consistent with Figs. 10(b) and 11.

   Fig. S2 shows the anomaly of the zonal mean oceanic potential temperature and salinity of 6ka and 127ka relative to PI. Surface cooling is consistent with Fig. 12 and freshening would be result of a more active water circulation in the NH. Strong freshening around 32°N in Fig. S2(f) is attributed to low salinity in the Mediterranean Sea.

## 4.4 Last Millennium and historical transient variabilities

   Figure 13(a) shows the time series of annual mean NH air temperature of LM and HIST. Sharp cooling events are clear responses to huge volcanic eruptions. The effect of solar forcing on annual mean temperature is unclear, probably because the signals are small in comparison with internal variability.

   The LIA is reasonably well expressed in the NH mean, but the warming during the Medieval Climate Anomaly (MCA; 950–
1250 CE), which is suggested by proxy data, is underestimated by the model. The difference between the NH mean temperature of the LIA and that of the MCA is −0.1 °C, which is not statistically significant in the Student's t-test.

   The HIST experiment was run for the period between 1850 and 2014 CE. Figure 13(b) shows the output of 30 ensembles of MIROC-ES2L historical experiments submitted to CMIP6 and data from HadCRUT4 (Morice et al., 2012), which is scaled at the mean value of 1960–1989. Centennial variabilities of the NH mean temperature in HIST and in the CMIP6 historical
experiments consistently show a positive trend during the first half of the 20[th] century, followed by a cooling trend until 1970

and then subsequent warming. In comparison with the standard historical experiments, HIST has a less positive bias (Fig. 13(c)).

## 5. Outlook and conclusions

Using MIROC-ES2L, an ESM that has recently been developed for CMIP6, we performed numerical experiments to examine the paleoclimate during several time periods and one historical experiment that was initiated from the LM experiment.

    Globally, there was reasonable agreement between the climate states described by the model and those derived from proxy data, while some regional discrepancies remained. In this section, we summarize the results and explore the possible causes of the discrepancies.

From PI, LGM conditions were introduced step by step into MIROC-ES2L and the LGM spin-up experiment was run for approximately 9,000 model years. The temperature anomaly of LGM relative to PI over the tropics is negative and there is general quantitative agreement between the anomaly derived from the model and that from proxy data (Bartlein et al., 2011; MARGO project members, 2009). This could be useful in constraining future projections, given that Annan et al. (2005), and Renoult et al. (2020) revealed the correlation between tropical cooling at LGM and ECS. It has been pointed out in the United

Nations Intergovernmental Panel on Climate Change (2013) Fifth Assessment Report that the cooling over Greenland during the LGM relative to PI is underestimated in the models. This could be attributed to strong AMOC in the models, which leads to an estimate of sea ice expansion over the northern Atlantic Ocean that is lower than suggested by proxy data. The anticorrelation between sea ice expansion and AMOC is known from observations (Boehm et al. 2015) and modeling (Peltier and Vettretti, 2014). Intrusion of Antarctic bottom water into the Atlantic Basin is very weak in MIROC-ES2L, even in the PI

experiment (Tatebe et al., 2018). Insufficient abyssal flow into the Atlantic Basin could be partly caused by the low resolution of the ocean component. Detailed analyses on the representation of atmospheric circulations would be necessary for further investigation. Model representation of the Southern Ocean might influence the distribution of $CO_2$ between the atmosphere and the ocean (Moore et al., 2000). Anomalies associated with topography might be obscured by the low horizontal resolution of the model, resulting in discrepancies between climate states in the model and those derived from proxy data. Cooling of

Eastern Antarctica during the LGM relative to PI, which is suggested by ice core data (−7 to −10 °C), is underestimated by this model (−6 °C), as explained in Sect. 4.2. This could be partly attributed to the positive SST bias over the Southern Ocean in the model at PI and subsequent underestimation of sea ice expansion. PMIP model analyses (Otto-Bliesner et al. 2007, Marozzochi and Jansen 2017) also suggested the correlation of AMOC and sea ice coverage.

    There is reasonable agreement between dust flux from the LGM experiment and that suggested by proxy data. However, the

PI experiment overestimates dust emissions from South America. Thus, the change in dust emission between LGM and PI is likely to be underestimated in the model, leading to underestimates of LGM anomalies relative to PI for climate (Ohgaito et

al., 2018) and ecosystem activity in the Southern Ocean (Yamamoto et al., 2019). Further studies will be necessary to investigate the impact of representing reasonable dust emission and loading on climate.

We prescribed conventional land PFTs in the LGM experiment, as discussed in Sect. 3.2. In models that comprise a coupled dynamic vegetation model, climate states would be altered through biophysical feedback because of changes in vegetation cover (O'ishi and Abe-Ouchi, 2013).

The LGM experiment was also performed in the previous phase of PMIP using MIROC-ESM (Sueyoshi et al., 2013), which is the previous version of MIROC-ES2L. Owing to differences in forcing (mainly in terms of GHGs and ice sheets) and spin-up procedures, we are unable to make direct comparisons of the experiments conducted using the two versions of the model. However, there is a general tendency of the PMIP4 model to simulate less cooling at LGM relative to PI, which is a tendency that was also identified by Kageyama et al. (2020) in their comparison of LGM experiments from different versions of PMIP. Further sensitivity experiments using different boundary conditions could be helpful for identifying causes of this discrepancy. Although we conducted long spin-up for the LGM experiment, the abyssal salinity and oceanic temperature are not representative of the structure suggested by proxy data. This discrepancy might reflect model biases, e.g., SST bias, and/or difficulty in representing the AMOC state of the LGM.

The two interglacial experiments (6ka and 127ka) include different orbital parameters and GHG levels, and had long spin-up times that exceeded 1,400 years. Results showed warming over NH continents during boreal summer relative to PI, consistent with the direction of change suggested by proxy archives (Bartlein et al., 2011; Turney and Jones, 2010); however, the model underestimates the amount of warming. The discrepancy could be reduced by improving the experimental setup, such as replacing the prescription of PFTs by a process that could produce PFTs that are closer to the real conditions of the periods. The vegetation coupling greatly improves the representation of the warmings shown by proxies at the Arctic Ocean margin (O'ishi and Abe-Ouchi, 2011, O'ishi et al. in press CP). On the other hand, some inconsistency remains in inland areas such as inner Eurasia. Pfeiffer and Lohmann (2016) suggested that we need to take into account the uncertainty of the times of the proxy data.

Compared with PI, temperature over the tropics is generally lower in the 6ka experiment, which may partly be inconsistent with proxy data archives (Bartlein et al., 2011, Kaufman et al., 2020). Because the uncertainty of the marine proxies is so large (Hessler et al. , 2014) that it is unlikely to be possible to identify positive or negative SST anomalies at 6ka, we exclude them from our assessment here. However, cooling over the tropics could be considered a reasonable and direct response to net negative solar forcing. Thus, the discrepancy between the model and proxy data suggests that feedbacks that might play a role in the modeling of climate change are missing in the current model. Further improvement and expansion of the model would be necessary.

The precipitation anomaly shows a northward shift of peak precipitation in boreal summer in the NH. The precipitation anomaly over the Sahara Desert in the model is still smaller than that suggested by the proxy data archive, which is a mismatch that has been persistent throughout the long history of PMIP. It might be necessary to include new processes to maintain higher soil moisture in the interglacials (Hopcroft et al., 2017).

The LM experiment performed in this study showed clear responses of global temperature to huge volcanic eruptions, while the responses of global temperature to other forcings were unclear. Responses to external forcings except volcanos are likely to be small in comparison with internal variabilities.

The difference between model NH mean temperature at the LIA and that at the MCA is too small to be statistically significant. However, earlier studies suggested that signals might be more pronounced at regional scales (PAGES2k, 2015; Fernandez-Donado et al., 2013); thus, further investigations regarding regional scales would be necessary. The HIST experiment was initiated from the end of the LM experiment and it produced time series of global temperatures that are similar to those from the other historical experiments that were initiated from PI (Hajima et al., 2020). This suggests that the initial conditions used for the standard historical experiment in CMIP6 are appropriate for the simulation of global temperatures in the industrial era. Sensitivity experiments using different boundary conditions would be useful for identifying causalities to obtain more details in future analyses.

**Appendix: How we overcame the difficulties of the LGM experiment**

Various difficulties can be expected in the realization of the LGM experiment. We encountered the most difficulty when we incorporated the LGM conditions step-by-step during the LGM spin-up.

Figure A1 shows the evolution of sea ice thickness on the north coast of North America (averaged over 150°-180° E, 70° -75° N averaged for January to March) for the first half of the LGM spin-up. We lowered the GHG levels in step (1) in Fig. 5 and we changed the bathymetry and ice sheet grids (albedo) in step (2). The AMOC settled down after the shock spike, but sea ice built up to 50 m on one grid of the north coast of North America at the $3305^{th}$ year in Figure A1, which is the limit of acceptable thickness in the model and thus the experiment was unable to continue (Fig. A1). After various trials and errors, the introduction of the LGM elevation (step 3) changed the atmospheric circulation field and prevented the sea ice from being pushed to the north coast of North America. Thus, the sea ice thickness settled within a range that allowed the experiment to continue. We do not know whether this happens in other models, but we release this information for reference in case other studies find continuation of the LGM experiment impossible.

**Data availability**

The source code of MIROC-ES2L can be obtained from https://zenodo.org/record/3893386#.XuW9icvnhaQ. The source codes of the analyses and required input data can be found at https://zenodo.org/record/3893403#.XuY5CcvnhaQ. The DOIs of the experiments are listed in Table 1. The DOI for the historical experiments is 10.22033/ESGF/CMIP6.5602. The model outputs derived in this study are freely available through the Earth System Grid Federation (ESGF). Details regarding the

385 ESGF can be found on the website of the CMIP Panel (https://www.wcrp-climate.org/wgcm-cmip/wgcm-cmip6, last access: 10 November 2020).

**Author contributions**

RuO coordinated, prepared the boundaries, conducted the LGM, 6ka, and LM experiments, analyzed the results, and wrote the manuscript. AY conducted offline spin-up of the ocean ecosystem experiments for LGM, analyzed

390 the results, and wrote the manuscript. TH developed and provided MIROC-ES2L and the offline land ecosystem model, and advised on conducting the experiments. RyO conducted the 127ka experiment and provided code for the calendar adjustment. MA prepared most of the boundary conditions of the LM experiment and submitted the data to the ESGF. HT helped prepare the ocean mask for the LGM experiment. All authors contributed to discussions and to improvement of the manuscript.

395

**Competing interests**

The authors declare no competing interests.

**Acknowledgments**

This work was supported by the TOUGOU project "Integrated Research Program for Advancing Climate Models"

400 (grant number: JPMXD0717935715) of Ministry of Education, Culture, Sports, Science, and Technology of Japan (MEXT). The authors acknowledge funding from MEXT KAKENHI (grant numbers: 17H06323, 17H06104) and JAMSTEC for use of the Earth Simulator supercomputer. We would like to thank Qiong Zhang and an anonymous reviewer for their suggestions to greatly improve this work. We thank Osamu Arakawa for his support regarding data management and submission of data to the ESGF. We thank Dai Yamazaki and Takashi Obase for providing

405 the river and height data for the LGM. We thank the TOUGOU-b team for discussions and advice regarding the experiments. We thank Tina Tin PhD and James Buxton MSc from Edanz Group (https://en-author-services.edanzgroup.com/), for editing a draft of this manuscript.

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

Table 1. Settings for the experiments

| Experiment short name | PI | LGM | 6ka | 127ka | LM | HIST |
|---|---|---|---|---|---|---|
| Time interval | Preindustrial control (1850 CE) | Last Glacial Maximum (21,000 years before present) | 6,000 years before present | 127,000 years before present | 850–1849 CE | 1850-2014 CE |
| Greenhouse gas levels | $CO_2$ (ppm) 284.725  $N_2O$ (ppb) 273.02  $CH_4$ (ppb) 808.25 | 190  200  375 | 264.4  262  597 | 275  255  685 | Meinshousen et al. (2017) | Time varying observation Eyring et al. (2016) |
| Orbital parameters | Eccentricity 0.01672 Obliquity 23.45 Angular precession 102.04 | 0.018994 22.949 114.42 | 0.018682 24.105 0.87 | 0.039378 24.04 275.41 | Berger (1978), Schmidt et al. (2011) | Same as PI |
| Altitude | Present-day | ICE-6G_C | Same as PI | Same as PI | Same as PI | Same as PI |
| Dust | Calculated in the model | Calculated in the model with additional erodibility map | Same as PI | Same as PI | Same as PI | Same as PI |

| Ice sheets and land–sea distribution | Present-day | ICE-6G_C | Same as PI | Same as PI | Same as PI | Same as PI |
|---|---|---|---|---|---|---|
| DOI | 10.22033/ESGF/CMIP6.5710 | 10.22033/ESGF/CMIP6.5644 | 10.22033/ESGF/CMIP6.5646 | 10.22033/ESGF/CMIP6.5645 | 10.22033/ESGF/CMIP6.5666 | 10.22033/ESGF/CMIP6.5602 |

Table 2. Changes in dissolved inorganic carbon, dissolved oxygen, and water-mass age from PI referred to the global ocean inventory or global ocean mean values. Values in brackets are the PI results. Upper (Deep) ocean is above (below) 1000 m depth. Atmospheric $CO_2$ concentration is prescribed in each experiment.

| Experiment | Atmospheric $CO_2$ (ppm) prescribed | Export production (Pg C yr$^{-1}$) | $\Delta$DIC (Pg C) | | | $\Delta$Oxygen (mmol m$^{-3}$) | | | $\Delta$Age (yr) | | |
|---|---|---|---|---|---|---|---|---|---|---|---|
| | | | Global | Upper | Deep | Global | Upper | Deep | Global | Upper | Deep |
| PI | 284.725 | 7.92 | (36615) | (8603) | (28012) | (191) | (176) | (196) | (570) | (308) | (655) |
| LGM | 190 | 8.48 | –629 | –239 | –390 | +10 | +10 | +10 | –52 | –33 | –59 |

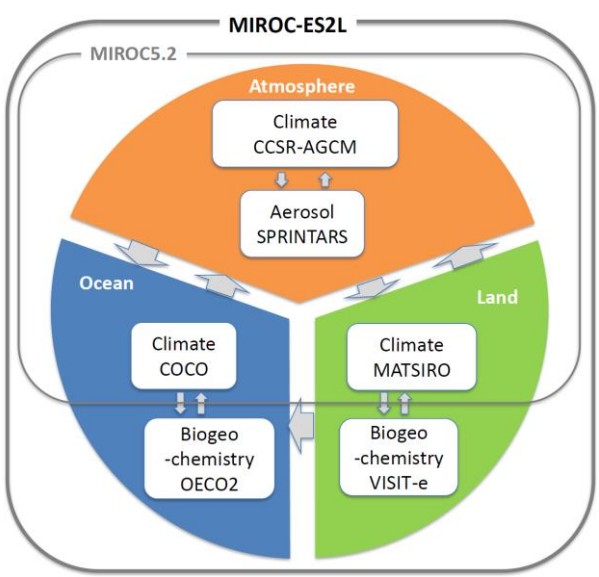

Figure 1: Schematic of MIROC-ES2L.

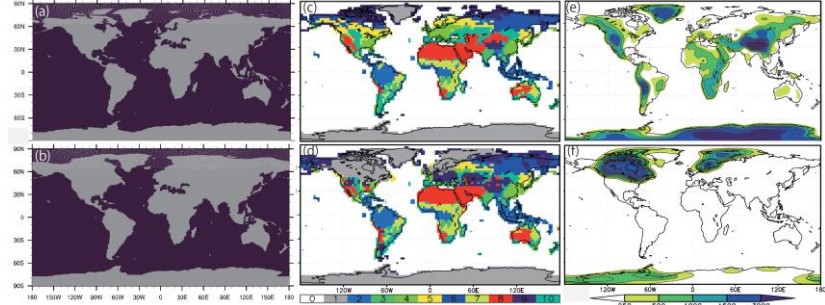

**Figure 2: Left panels: Land–sea distribution converted to 1° × 1° ocean grids for (a) PI, 6ka, and 127ka and (b) LGM. Middle panels: Distribution of land vegetation types for (c) PI, 6ka, and 127ka and (d) LGM. Numbers in color bar represent vegetation types: 1) ice sheets, 2) broadleaf evergreen forest, 3) broadleaf deciduous forest and woodland, 4) mixed coniferous and broadleaf deciduous forest and woodland, 5) coniferous forest and woodland, 6) high-latitude deciduous forest and woodland, 7) wooded C4 grassland, 8) shrubs and bare ground, 9) tundra, and 10) C3 grassland. Right panels: (e) Altitude for PI, 6ka, and 127ka and LM (unit: m) and (f) altitude anomaly (unit: m) given for the LGM experiment based on ICE-6G_C (Peltier, 2015).**

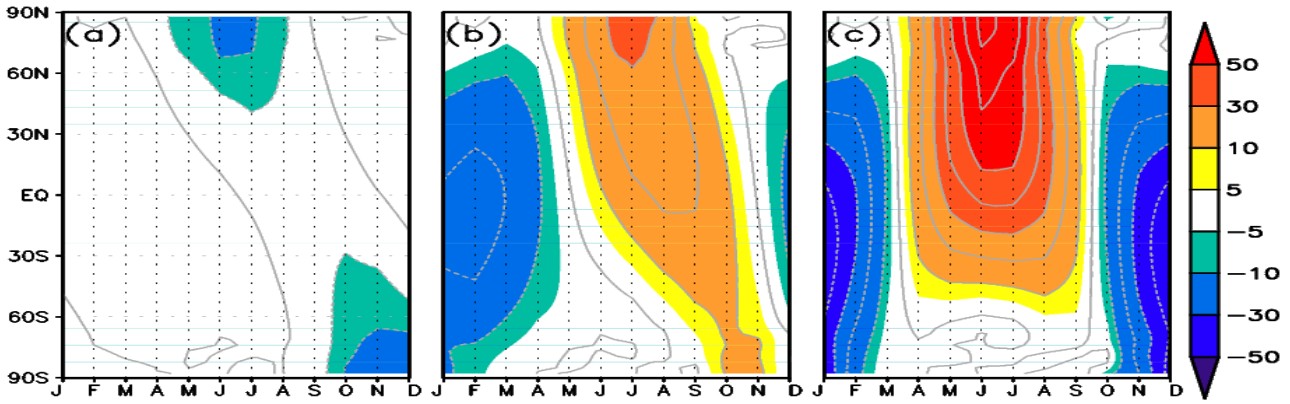

**Figure 3: Variation of incoming shortwave solar radiation anomaly relative to PI (unit: W m⁻²) with season and latitude for (a) LGM, (b) 6ka, and (c) 127ka.**

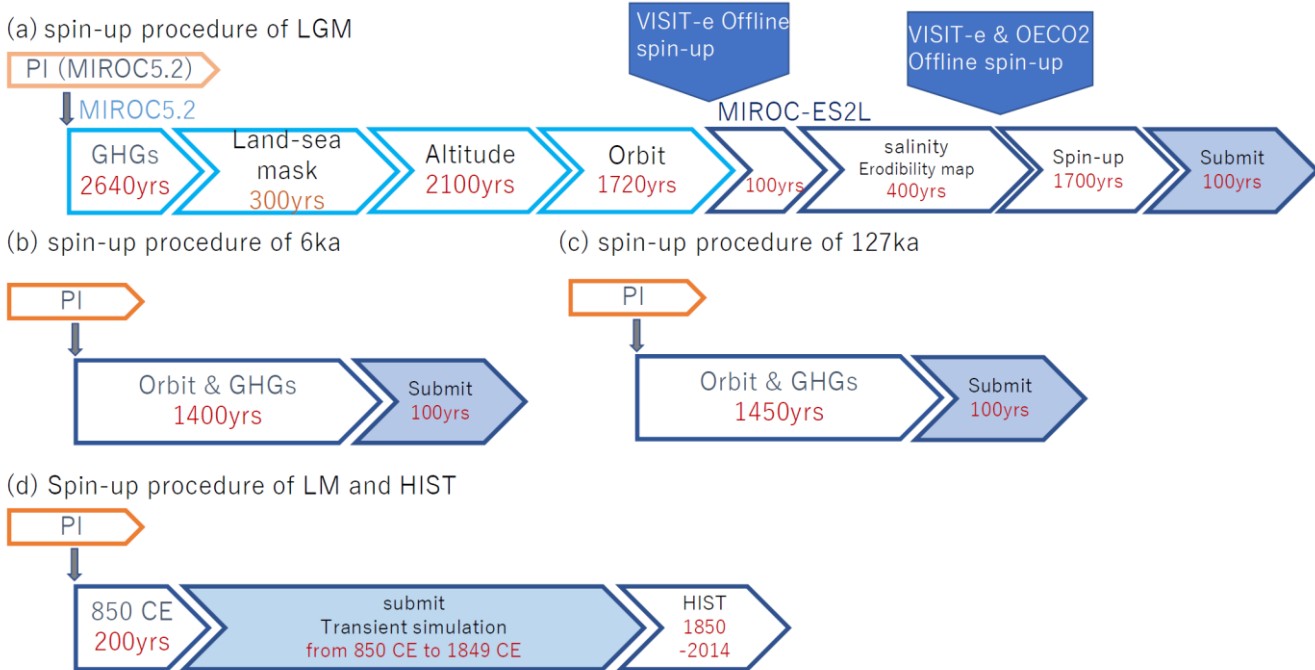

**Figure 4: Schematic showing spin-up procedures of the following experiments: (a) LGM, (b) 6ka, (c) 127ka, and (d) LM and HIST. Shapes with dark orange or dark blue outlines represent experiments using MIROC-ES2L. Shapes with light orange or light blue outlines represent experiments using MIROC5.2. Shapes filled in pale blue represent model output submitted to PMIP4–CMIP6.**


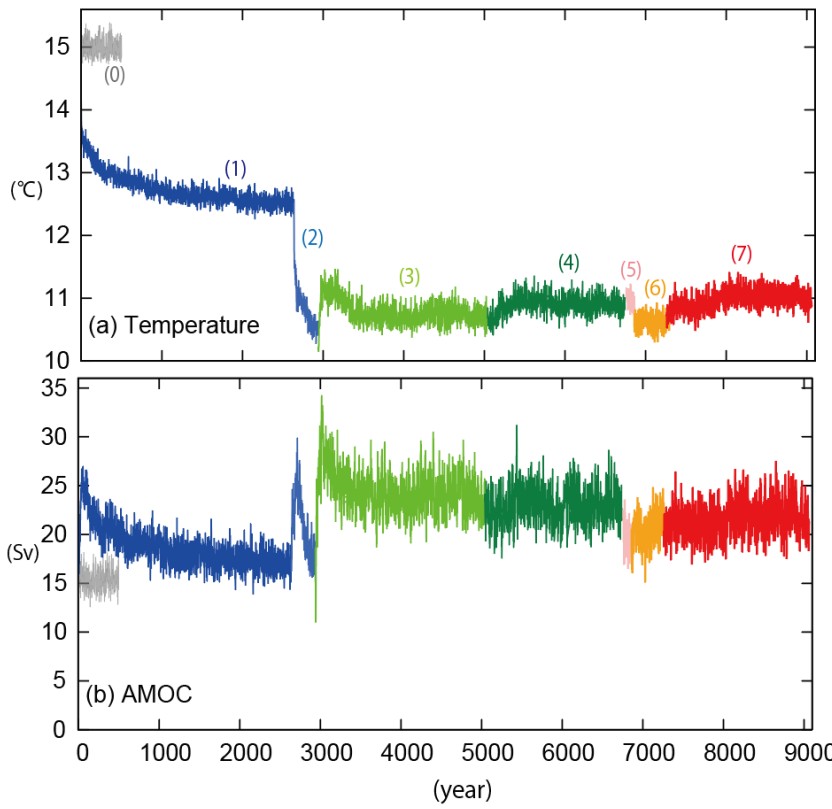

**Figure 5: Time series for spin-up and submitted period (final 100 years) of LGM experiment and PI as a reference for (a) global mean air temperature at 2 m height and (b) peak values of annual mean AMOC. Gray line (0) denotes PI value. Blue line (1): the experiment with only GHG levels set to LGM values. Light blue line (2): with the land–sea distribution and land PFTs hanged to the LGM states. Light green line (3): with altitude set to the LGM state. Dark green line (4): with orbit of the Earth set to the LGM value. Pink line (5): spin-up experiment using MIROC-ES2L after VISIT-e offline spin-up. Orange line (6): with erodibility map and offset of ocean salinity applied. Red line (7): final spin-up after offline spin-up experiments by VISIT-e and OECO2.**

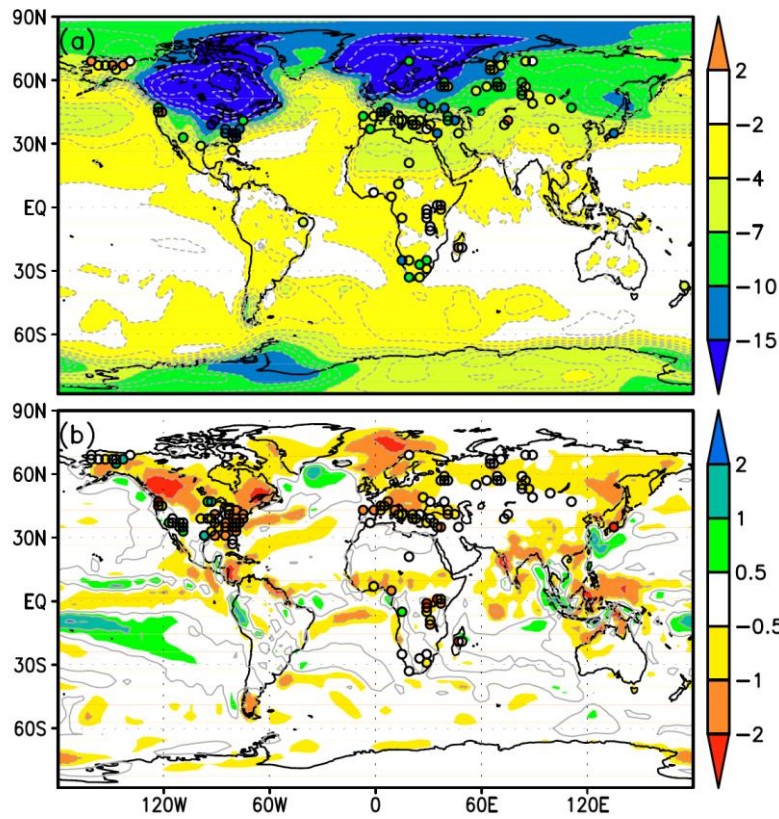


**Figure 6: (a) Air temperature anomaly at 2 m height (unit: °C) and (b) precipitation anomaly (unit: mm d⁻¹). Anomalies are calculated as LGM relative to PI values. Circles denote values derived from proxy data (Bartlein et al., 2011).**

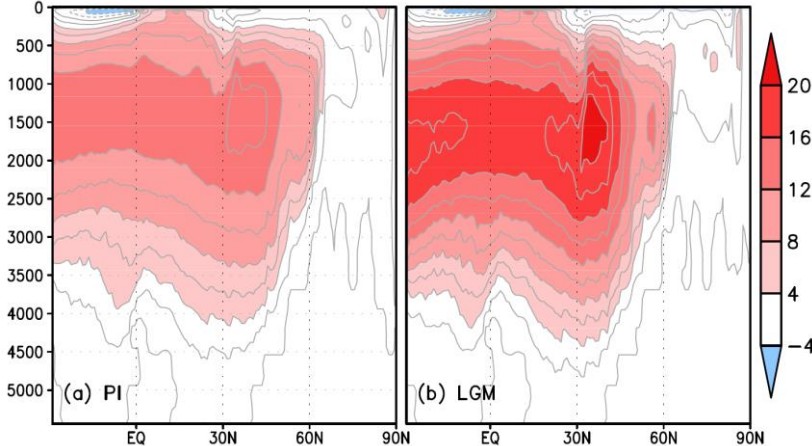

**Figure 7: Meridional overturning streamfunction for the Atlantic Basin (unit: Sv) for (a) PI and (b) LGM.**

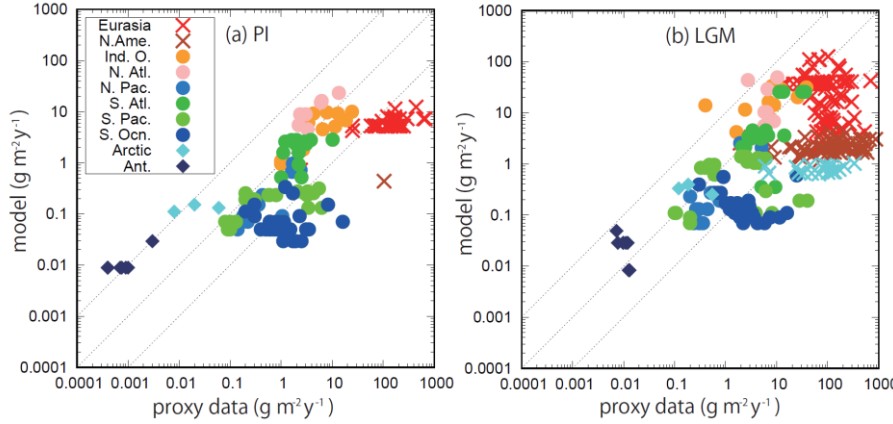


**Figure 8: Dust deposition from model output and derived from proxy archives (Kohfeld et al., 2013; Albani et al., 2014) for (a) PI and (b) LGM (g m⁻² yr⁻¹).** Colors represent the locations of the proxy data, as explained in the legend in the figure. Crosses, circles, and diamonds represent terrestrial, marine, and ice core data.

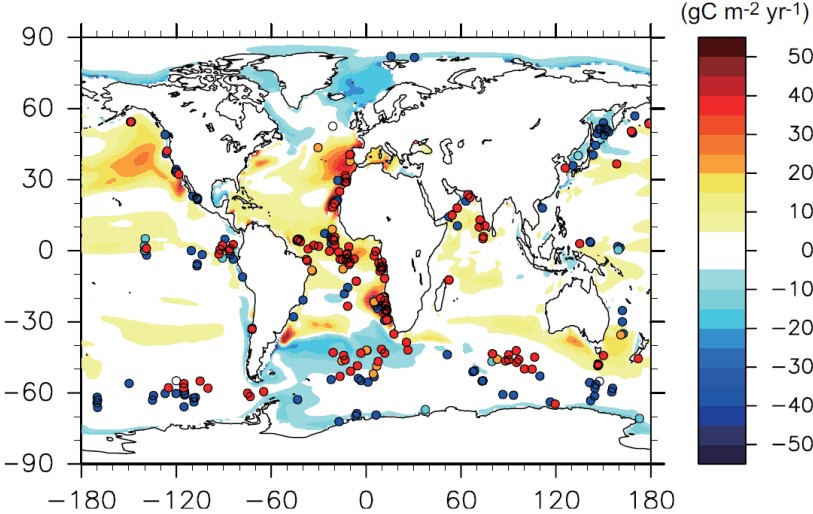


**Figure 9: Primary production anomaly of the oceanic ecosystem (unit: gC m⁻² yr⁻¹).** Anomalies are calculated as LGM relative to PI values. Circles denote qualitative changes in primary production derived from proxy data (Kohfeld et al., 2013).

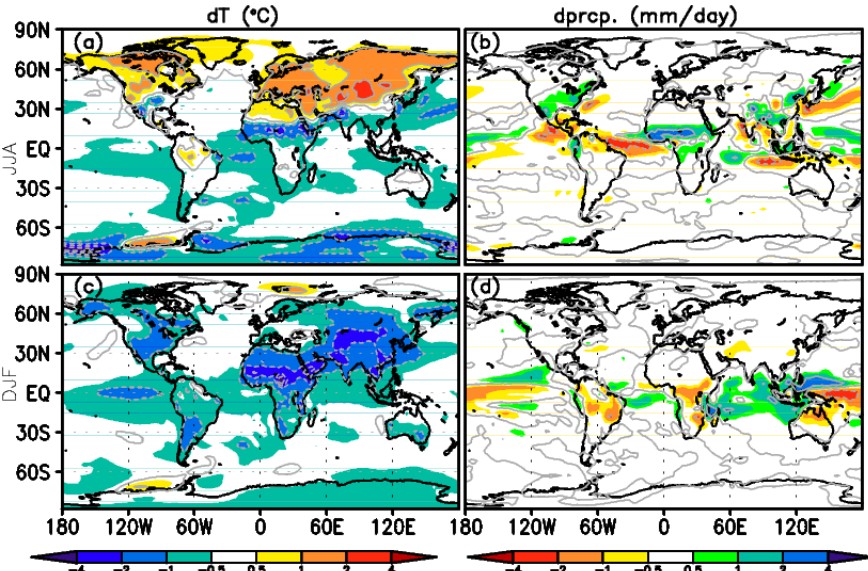

**Figure 10:** Seasonal temperature anomaly for (a) JJA and (c) DJF. Seasonal precipitation anomaly for (b) JJA and (d) DJF. Anomalies are calculated as 6ka relative to PI values. Calendar adjustments are applied.

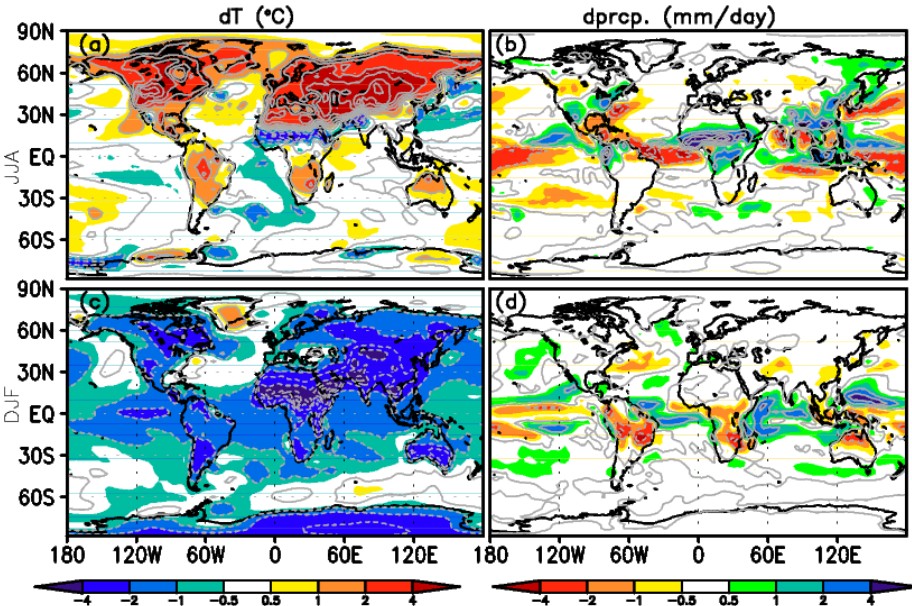

**Figure 11:** Same as Fig. 10 but for 127ka.

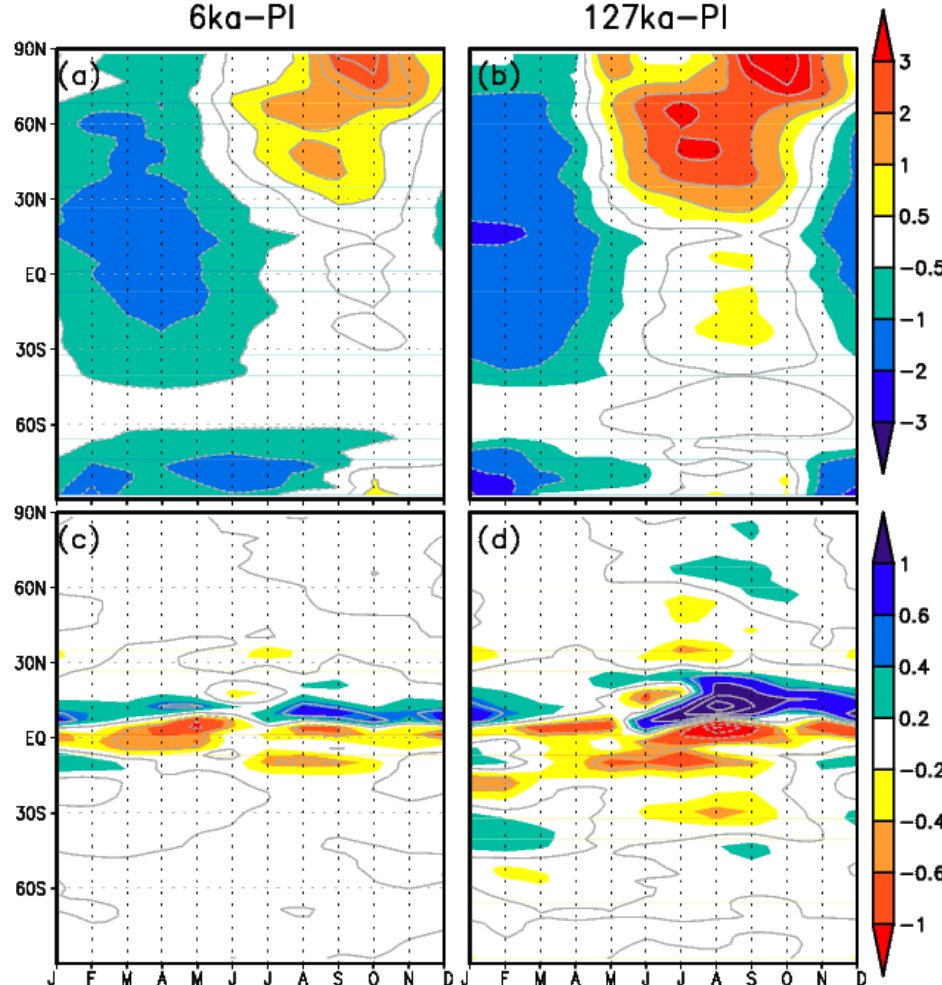


**Figure 12: Hovmöller diagrams for (a) air temperature anomaly at 2 m height (°C) for 6ka relative to PI, (b) air temperature anomaly at 2 m height (°C) for 127ka relative to PI, (c) precipitation anomaly (mm d$^{-1}$) for 6ka relative to PI, and (d) precipitation anomaly (mm d$^{-1}$) for 127ka relative to PI. Calendar adjustments are applied.**

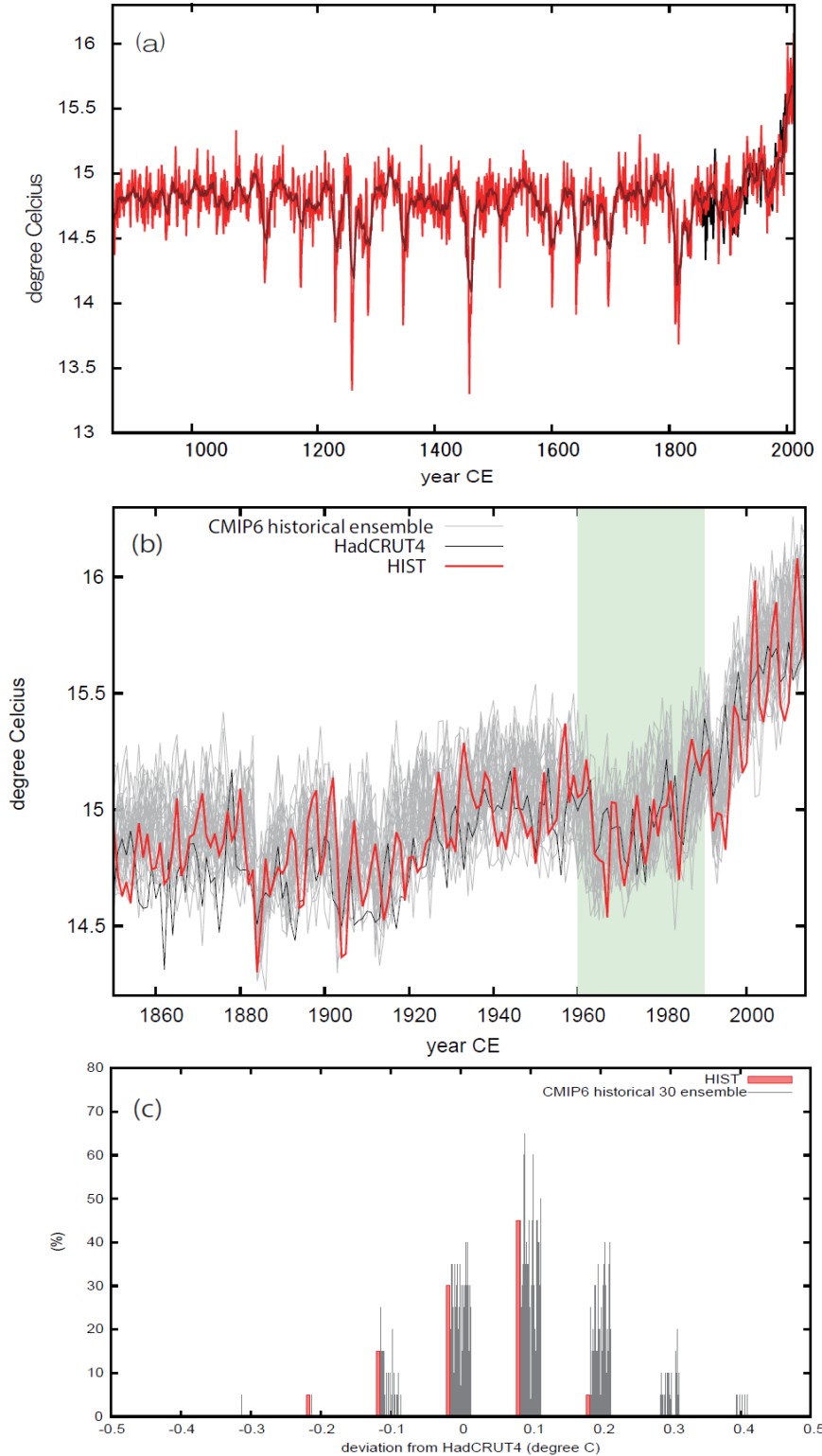

**Figure 13: (a) Annual mean air temperature (°C) averaged over the Northern Hemisphere from the LM and HIST experiments (red: annual mean, dark red: 10 year running mean) and observational data from HadCRUT4 (black). (b) Annual mean air temperature (°C) averaged over the Northern Hemisphere from 1850–2014 for HIST and 30 ensemble members of the historical experiments (gray) and HadCRUT4 (black). HadCRUT4 is scaled for the period from 1961–1990 (period shaded light green). (c)**
**Histogram of deviations from HadCRUT4 shown in (b), averaged for every five-year mean during 1850–1949, counted in 0.1 °C increments in bins. Red: HIST experiment, Gray: CMIP6 historical ensemble of 30 members.**

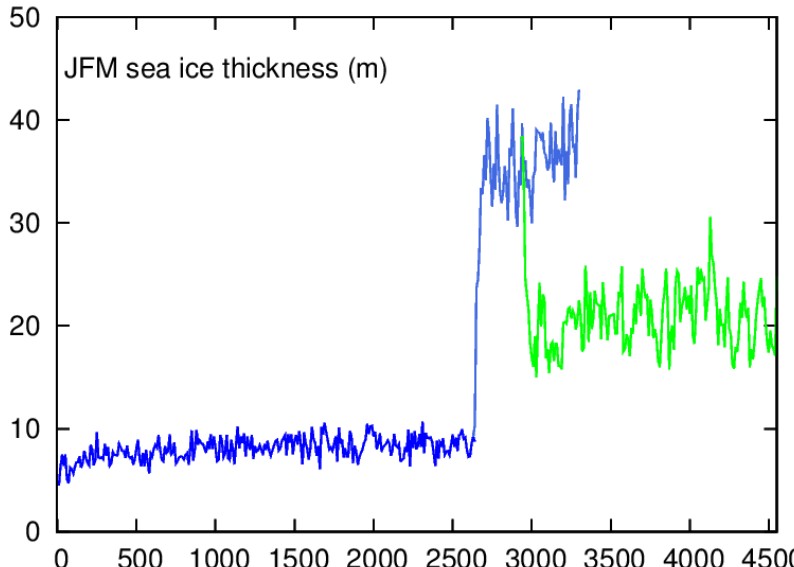

**Figure A1: Average sea ice thickness (m) from January to March for the first half of the spin-up of the LGM experiment, averaged**
**150° - 180° E, 70° -75° N. The blue line: step 1, light blue: step 2, and yellow-green: step 3 in Figure 5, respectively. For simplicity, data for one year for every 10 years are plotted.**