# Peer review of "PMIP4 experiments using MIROC-ES2L Earth System Model"

_Geoscientific Model Development, 2020_

## Referee Comment (RC1) · Anonymous Referee #1 · 11 Aug 2020

The paper summarizes PMIP4 experiments using the Model for Interdisciplinary Research on Climate Earth System Model (MIROC-ES2L). Experiments for PI, LGM, interglacials (6k, 127k), LM and historical are presented. The MIROC-ES2L is an ESM developed for CMIP6 (Tatebe et al. 2018, Scientific Reports; Hajima et al. 2020, GMD), but the version has more ESM components like the ecosystem, aerosol and vegetation modules. Most analyses are however related to the more standard physical quantities like SAT, precipitation, and ocean circulation (AMOC).

The paper needs some revisions before publication, somehow in between minor and major revisions. Part of the analysis is not very deep and a little speculative, some innovative aspects of the new model as the ocean biogeochemical model OECO2 are not considered in detail. A positive aspect of the paper is the compilation of different PMIP

experiments in one paper. The evaluation of the climate sensitivity is not mentioned.

Here are specific critics, some of them are only minor:

1) page 2, line 46, Because cooling at LGM relative to PI is at a comparable level to present-day global warming,

-this statement is not valid. The present day warming with respect to PI is in the order of 0.5-1 K, the cooling LGM-PI is in the order of 3 K, regionally much larger (e.g. 10 K or more)

2) page 3, line 68, However, models have been unable to reproduce the quantitative changes recorded in proxy data.

-Please provide a reference. This statement is not very specific. Please modify and be explicit saying which type of paleoclimate data you are refering to.

3) page 4, SECTIONS 3.2 and 3.3 setup and spin-up:

-Specify how you treat the PFTS. It is not mentioned in the text, but shown in Fig. 4

4) page 8, line 252 We prescribed conventional land PFTs in the LGM experiment.

-This is not clear. The reader thinks that all experiments work with prescribed PFTs.

5) The language needs some improvements.

6) page 7, line 199: calculated for June to August (JJA) and December to 200 February (DJF).

-Please discuss the seasonality issue for past climates. Similar isse in Fig. 12: Please correct for the paleo-calendar (e.g. following Braconnot)

7) page 8, line 223: There is also good agreement between HadCRUT4 data and output from all of the historical experiments at the multi-decadal time scale.

-Be more specific, "good agreement" can be substanciated

8) page 8, line 237 This could be attributed to a strong AMOC in the models, which leads to an estimate of sea ice expansion over the northern Atlantic Ocean that is lower than that suggested by proxy data.

-a strong AMOC would reduce the sea ice? please comment

9) page 8, line 240 Positive SST bias over the Southern Ocean in the model at PI may also contribute towards the underestimation of abyssal flow and could result in a persistently strong AMOC at LGM.

-too speculative, please substanciate your statement

10) page 8, line 245 Cooling of Eastern Antarctica at LGM relative to PI that is suggested by ice core data is underestimated by the model.

-please provide references and numbers.

11) page 9, line 263 This is consistent with the direction of change suggested by proxy archives (Bartlein et al., 2011; Turney and Jones, 2010)

-Be aware of the proxy for temperature during LIG, it is related to peak interglacial conditions. See e.g. Pfeiffer and Lohmann (2016, CP) for a discussion on that.

12) page 9, line 269 the degree of improvement would be area dependent.

-please be more specific, too vague

13) page 9, line 269 Compared with PI, temperature over the tropics is lower in the 6ka experiment, which contradicts with proxy data.

-This is not correct, see, e.g. Lohmann et al. (2013, CP) for the SST data and model-data comparisons

14) page 24, line 674, peak values of annual mean AMOC.

-please exclude the surface layers since they reflect the wind-driven part. In several papers, the upper 300 m (or similar) are excluded.

15) page 25, caption of Fig 5:

-the colors are partly difficult to identify, e.g. light blue.

16) LGM: in the paper, please mention the potential bias due to the choice of initial condition. E.g. the deep ocean salinity structure is quite different from the modern one. It shall be mentioned that the spin up procedure, the initial condition, and the limited sin up time of less that 2000 years might be related to this mismatch.

17) page 25, AMOC plots: the figures shall be improved by inserting the minimum ocean depth (e.g. in grey)

18) Figure 10: Indeed a week precipitation response in the tropics and subtropics. Is the zonal water vapor transport too small ?

19) Please mention the model's climate (or ES) sensitivity in the paper.

---

## Referee Comment (RC2) · Anonymous Referee #2 · 14 Aug 2020

The manuscript documents four PMIP4 experiments setup with MIROC-ES2L Earth system model, and evaluate the model performance by comparing with the published proxy data indication. The authors made efforts to run long spin-up for LGM and presented the spin-up process step by step in detail. The other three experiments setup are relatively easier to setup and needs shorter spin-up time than the LGM experiment. The evaluation of the model results are shown for temperature and precipitation through model-data comparison, which is understandable since only these climate parameters are widely reconstructed. MIROC-ES2L is an earth system model, and most of the components are turned on for the PMIP4 experiments (my guess, the authors should confirm this in the paper), means the model is able to produce more physical parameters than those available from proxy data. It is worthy to present more features

such as sea-ice, deep ocean temperature and salinity, carbon cycle, modelled dust etc, to show the advantages of an earth system model. I suggest the authors do a major revision by adding more information to promote the ESM's capability.

Specific comments:

Line 53-54: Are these models include the interactive dust, or do you mean the pre-scribed dust emission is not proper and may influence the simulated temperature? It would be interesting to see the dust simulated in MIROC-ES2L and compare with the prescribed dust, especially for LGM.

Line 98: "The ecosystem modules can simulate global carbon and nitrogen cycles ex-plicitly." As listed in table 1 for all the experiments the GHG concentrations are following the PMIP4 protocol. It is not clear if the ecosystem modules are not turned on and how does the model treat the CO2 and N2O in the atmosphere, please clarify.

Line 100: "Dynamics of aerosols are calculated by an online aerosol module". Since most model that does not have an interactive aerosol module use the prescribed PI aerosol for all the past periods, I am curious if the dynamical module in MIROC-ES2L simulated aerosols, such as dust, are different from those prescribed aerosols.

Line 105-106: Are the model configurations (interactive components) and resolutions same in the DECK and PMIP4 experiments?

Line 138-140: These parameters are listed in the table 1 and no need to repeat in the text.

Page 21, table 2: This table does not provide more information than the description in the text, either remove this table or provide more specific information than only given the reference.

Line 680, Fig6b: there is a sharp gradient at around 30N, can you explain?

Line 221-225, regarding the HIST part in Fig13, more information about the three ensembles during HIST period are needed. The HIST part in Fig13 is hard to observe and compare. It would be more informative to show another figure only for HIST part, in order to draw the conclusion that the initial conditions for HIST from the end of LM experiment is similar to that from the long PI run, and discuss if this is the case for other models or it might be model dependent.

The authors present the four experiments separately, a summary table or figure to compare the four past periods would be helpful to have an overview of the climate change, and differences of modelled glacial and interglacial climate.

Minors:

Line 36, "the Pliocene", should be " mid-Pliocene (3.2 million years before present)". Line 181, "by PI", suggest change to "in PI or at PI".

---

## Author Comment (AC1) · 9 Nov 2020

**Reply for anonymous reviewer #1 of PMIP4 experiments using MIROC-ES2L Earth System Model**

Rumi Ohgaito, Akitomo Yamamoto, Tomohiro Hajima, Ryouta O'ishi, Manabu Abe, Hiroaki Tatebe, Ayako Abe-Ouchi, Michio Kawamiya

**Thank you, the anonymous reviewer, for the thought-provoking and constructive comments. In the following reply, the reviewer's comments are written in black texts and our responses are in **bold** and **blue** texts.**

The paper summarizes PMIP4 experiments using the Model for Interdisciplinary Research on Climate Earth System Model (MIROC-ES2L). Experiments for PI, LGM, interglacials (6k, 127k), LM and historical are presented. The MIROC-ES2L is an ESM developed for CMIP6 (Tatebe et al. 2018, Scientific Reports; Hajima et al. 2020, GMD), but the version has more ESM components like the ecosystem, aerosol and vegetation modules. Most analyses are however related to the more standard physical quantities like SAT, precipitation, and ocean circulation (AMOC). The paper needs some revisions before publication, somehow in between minor and major revisions. Part of the analysis is not very deep and a little speculative, some innovative aspects of the new model as the ocean biogeochemical model OECO2 are not considered in detail. A positive aspect of the paper is the compilation of different PMIP experiments in one paper. The evaluation of the climate sensitivity is not mentioned.

More earth system analysis has been augmented such as discussions on biogeochemical cycles at LGM, and revisions have been made to the text. In Addition, because we realized many modelling groups have difficulty in conducting LGM experiment, we added Appendix describing the most difficulty we encountered during the spin-up of the LGM experiment. Climate sensitivity has also been mentioned in Introduction, Sect. 2 and Sect. 5. We will respond to each comment below.

1) page 2, line 46, Because cooling at LGM relative to PI is at a comparable level to presentday global warming, -this statement is not valid. The present day warming with respect to PI is in the order of 0.5-1 K, the cooling LGM-PI is in the order of 3 K, regionally much larger (e.g. 10 K or more)

**"Present-day global warming" was misleading; this is a comparison between ECS and LGM-PI.**

These changes do not have to match exactly, but it is better to have some large changes in the recent past where the ocean-land distribution does not change much from the present

**day, which can be used to constrain the ECS (Annan et al. 2005, Renoult et al. 2020). We discussed this in the text.**

2) page 3, line 68, However, models have been unable to reproduce the quantitative changes recorded in proxy data. -Please provide a reference. This statement is not very specific. Please modify and be explicit saying which type of paleoclimate data you are referring to.

**We added McKay et al., 2011, Capron et al., 2014, Hoffman et al., 2017 in the manuscript.**

3) page 4, SECTIONS 3.2 and 3.3 setup and spin-up: -Specify how you treat the PFTS. It is not mentioned in the text, but shown in Fig. 4

4) page 8, line 252 We prescribed conventional land PFTs in the LGM experiment. -This is not clear. The reader thinks that all experiments work with prescribed PFTs.

As you pointed out, the explanation of the PFTs was insufficient. We have added the following explanation to text in Sect 3.1.

"The PFTs in PI are inherited from MIROC-ESM (Watanabe et al. 2011), which was based on Ramankutty and Foley (1999). "

The definition of the PFTs of LGM is also described in Sect 3.2 as follows.

"The LGM PFTs were created on the PI PFTs with the ice sheet grids defined by ICE-6G\_C, and nearby PFTs were diverted to non-ice sheet land (exposed continental shelves) that expanded from PI.", "The erodibility map specifies low latitudes as deserts and midto high latitudes as tundra"

5) The language needs some improvements.

**Language was improved by a professional language reviewer.**

6) page 7, line 199: calculated for June to August (JJA) and December to 200 February (DJF). -Please discuss the seasonality issue for past climates. Similar isse in Fig. 12: Please correct for the paleo-calendar (e.g. following Braconnot)

Calendar adjustments were introduced to LGM, 6ka, and 127ka, and the related figures were replaced.

7) page 8, line 223: There is also good agreement between HadCRUT4 data and output from all of the historical experiments at the multi-decadal time scale. -Be more specific, "good agreement" can be substanciated

After submitting this manuscript, we expanded the historical experiments for CMIP6, up to 30 ensemble members. The CMIP6 standard historical experiments were removed from Figure 13(a) because it is difficult to identify. Figure 13 (b) shows the HIST experiment starting from 1850 with standard 30 members and comparison with HadCRUT4, and (c) shows the histogram of biases from HadCRUT4 for the period from the late 19th century until the first half of the 20th century. The results showed that the HIST showed less positive bias than the standard historical experiments.

8) page 8, line 237 This could be attributed to a strong AMOC in the models, which leads to an estimate of sea ice expansion over the northern Atlantic Ocean that is lower than that suggested by proxy data. -a strong AMOC would reduce the sea ice? please comment

Correlation between strong AMOC and sea ice retreat has been reported from observation and modelling studies (Boehm et al. 2015, Peltier and Vettretti, 2014). This has been added to the text.

9) page 8, line 240 Positive SST bias over the Southern Ocean in the model at PI may also contribute towards the underestimation of abyssal flow and could result in a persistently strong AMOC at LGM. -too speculative, please substanciate your statement

As you pointed out, the statement was too speculative. It was changed as follows.

"Insufficient abyssal flow into the Atlantic Basin could be partly caused by the low resolution of the ocean component. Detailed analyses on the representation of atmospheric circulations would be necessary for further investigation. Model representation of the Southern Ocean might influence the distribution of  $CO_2$  between the atmosphere and the ocean (Moore et al., 2000). Anomalies associated with topography might be obscured by the low horizontal resolution of the model, resulting in discrepancies between climate states in the model and those derived from proxy data. Cooling of Eastern Antarctica during the LGM relative to PI, which is suggested by ice core data (-7 to -10 °C), is underestimated by this model (-6 °C), as explained in Sect. 4.2. This could be partly attributed to the positive SST bias over the Southern Ocean in the model at PI and subsequent underestimation of sea ice expansion. PMIP model analyses (Otto-Bliesner et al. 2007, Marozzochi and Jansen 2017) also suggested the correlation of AMOC and sea ice coverage."

10) page 8, line 245 Cooling of Eastern Antarctica at LGM relative to PI that is suggested by

ice core data is underestimated by the model. -please provide references and numbers.

**We rewrote as follows,**

Cooling of Eastern Antarctica at LGM relative to PI that is suggested by ice core data (-7 to -10 degree C (Stenni et al. 2010, Uemura 2012) is underestimated by the model (-5.1 degree C).

11) page 9, line 263 This is consistent with the direction of change suggested by proxy archives (Bartlein et al., 2011; Turney and Jones, 2010) -Be aware of the proxy for temperature during LIG, it is related to peak interglacial conditions. See e.g. Pfeiffer and Lohmann (2016, CP) for a discussion on that.

The following has been added to the text.

Pfeiffer and Lohmann 2016 suggested that we need to take into account the uncertainty of the times of the proxy data.

12) page 9, line 269 the degree of improvement would be area dependent. -please be more specific, too vague

Rewritten as follows. "The vegetation coupling greatly improves the representation of the warmings shown by proxies at the Arctic Ocean margin (O'ishi and Abe-Ouchi, 2011, O'ishi et al. in press CP). On the other hand, some inconsistency remains in inland areas such as inner Eurasia."

13) page 9, line 269 Compared with PI, temperature over the tropics is lower in the 6ka experiment, which contradicts with proxy data. -This is not correct, see, e.g. Lohmann et al. (2013, CP) for the SST data and modeldata comparisons

Thank you for letting us know Lohmann et al. (2013, CP). We changed the description to "Compared with PI, temperature over the tropics is lower in the 6ka experiment, which is in the range of variability of the proxy data (Bartlein et al. 2011, Lohmann et al. 2013). ".

14) page 24, line 674, peak values of annual mean AMOC. -please exclude the surface layers since they reflect the wind-driven part. In several papers, the upper 300 m (or similar) are excluded.

The peak value between 15 - 60 N and between 950-3300 m was taken as the peak value

of AMOC in the analyses. This is described at the end of Sect 4.1 in the text.

15) page 25, caption of Fig 5: -the colors are partly difficult to identify, e.g. light blue.

We put numbers in the figure to identify each experiment, and described them in the caption.

16) LGM: in the paper, please mention the potential bias due to the choice of initial condition. E.g. the deep ocean salinity structure is quite different from the modern one. It shall be mentioned that the spin up procedure, the initial condition, and the limited sin up time of less that 2000 years might be related to this mismatch.

The LGM spin-up was integrated for 6760 years using the physical core AOGCM to take longer, as described in Sect. 3.2, and 2200 years after adding the giogeochemical modulus (Figs. 4, 5). That is, we submitted 100 years after a total of 8960 years of spin-up as a physical field for temperature, salinity, etc. to CMIP6/PMIP4. This is sufficiently longer than the length of the deep ocean circulation.

The distribution of salinity and ocean temperature, as you pointed out, was also added to Supplemental Fig. S2 and described in text Sect. 4.2, 4.3 and discussed in Sect. 5.

17) page 25, AMOC plots: the figures shall be improved by inserting the minimum ocean depth (e.g. in grey)

We leave the figures as they are because the minimum ocean depth of the model is 1 m, it cannot be resolved in the figures of full ocean depth.

18) Figure 10: Indeed a week precipitation response in the tropics and subtropics. Is the zonal water vapor transport too small ?

The zonal water vapor transport is shown in Supplemental Fig. S1. The results show an overall decrease in water vapor transport in the PI. This is described in Sect. 4.2.

19) Please mention the model's climate (or ES) sensitivity in the paper.

ECS of MIROC-ES2L is 2.66. The relation between paleoclimate and climate sensitivity is added in the Introduction and the value is described in Sect. 2. Discussions are added in Sect 5.

---

## Author Comment (AC2) · 9 Nov 2020

**Reply for anonymous reviewer #2 of PMIP4 experiments using MIROC-ES2L Earth System Model**

Rumi Ohgaito, Akitomo Yamamoto, Tomohiro Hajima, Ryouta O'ishi, Manabu Abe, Hiroaki Tatebe, Ayako Abe-Ouchi, Michio Kawamiya

**Thank you, the anonymous reviewer, for the thought-provoking and constructive comments. In the following reply, the reviewer's comments are written in black texts and our responses are in bold and blue texts.**

The manuscript documents four PMIP4 experiments setup with MIROC-ES2L Earth system model, and evaluate the model performance by comparing with the published proxy data indication. The authors made efforts to run long spin-up for LGM and presented the spin-up process step by step in detail. The other three experiments setup are relatively easier to setup and needs shorter spin-up time than the LGM experiment. The evaluation of the model results are shown for temperature and precipitation through model-data comparison, which is understandable since only these climate parameters are widely reconstructed. MIROC-ES2L is an earth system model, and most of the components are turned on for the PMIP4 experiments (my guess, the authors should confirm this in the paper), means the model is able to produce more physical parameters than those available from proxy data. It is worthy to present more features such as sea-ice, deep ocean temperature and salinity, carbon cycle, modelled dust etc, to show the advantages of an earth system model. I suggest the authors do a major revision by adding more information to promote the ESM's capability.

**Thank you for properly evaluating our work. As you say, there are few analyses that take advantage of the properties of the Earth System Model, so we have compiled additional analyses discussing the biogeochemical cycles of LGM and revised the text accordingly.**
**The answers to specific comments will be given one by one in the following.**

 Specific comments:
Line 53-54: Are these models include the interactive dust, or do you mean the prescribed dust emission is not proper and may influence the simulated temperature? It would be interesting to see the dust simulated in MIROC-ES2L and compare with the prescribed dust, especially for LGM.

**It was poorly explained and misleading. We added the following in the relevant section.**
 **"The dust deposition was several to tens of times higher at LGM (Lambert et al. 2008, Lamy et al. 2014, Dome Fuji Ice Core Project members 2017), but was difficult to reproduce by LGM**

experiments; to reproduce the dust abundance at LGM, we need to assume glaciogenic dust (Mahowald et al. 2006, Ohgaito et al. 2018), or assuming an erodibility map (Albani et al. 2014). And an erodibility map was formally introduced in PMIP4 (Kageyama et al. 2017), in addition to the dust emission that is simulated in non-Paleo simulations. In Ohgaito 2018, they showed that simulated dust affects the temperature around Antarctica."

The simulated dust is shown in Fig. 8 and Supplemental Fig. S3 and explained in Sect. 4.2. We think that the sensitivity experiments that prescribe dust are interesting in assessing the impact of dust changes on climate, but it is beyond the scope of the description paper of PMIP4 experiments. We add discussion on it in Sect .5 as a suggestion for future study.

Line 98: "The ecosystem modules can simulate global carbon and nitrogen cycles explicitly." As listed in table 1 for all the experiments the GHG concentrations are following the PMIP4 protocol. It is not clear if the ecosystem modules are not turned on and how does the model treat the $CO_2$ and $N_2O$ in the atmosphere, please clarify.

The model itself calculates carbon and nitrogen fluxes by OECO2 and VISIT-e, but in these experiments, the simulated $CO_2$ and $N_2O$ fluxes do not change the atmospheric concentrations and thus their changes do not feedback on climate (i.e., the concentrations are prescribed to the PMIP4 specified values and the fluxes are simulated for the diagnostic purposes). This has been added to Sect 2, end of 1st paragraph.

Line 100: "Dynamics of aerosols are calculated by an online aerosol module". Since most model that does not have an interactive aerosol module use the prescribed PI aerosol for all the past periods, I am curious if the dynamical module in MIROC-ES2L simulated aerosols, such as dust, are different from those prescribed aerosols.

Aerosols are calculated online in the aerosol module SPRINTARS (Fig 1 and details at Takemura et al. 2000, 2002, 2005, and 2009). In the case of dust, the amount of dust generated is determined by the values of wind speed, soil moisture, vegetation type, snow cover, and LAI for each time step.
Figure 8 compares PI and LGM dust deposition to various proxy data archives. An additional comparison of the deposition maps and proxy archives is shown in Supplemental Fig. S3. An explanation of that figure is given in the text, Sect 4.2.

Line 105-106: Are the model configurations (interactive components) and resolutions same in the DECK and PMIP4 experiments?

Yes. These PMIP4 experiments use the same binary as the DECK experiments. That is, they have the same resolution and the same configurations.
The listed input data given in Table 1 are different from PI. The explanations are added in the manuscript at the end of Sect. 2.

Line 138-140: These parameters are listed in the table 1 and no need to repeat in the text.

The sentences have been changed to be more descriptive, such as "The main difference between these periods and the PI period was the change in insolation attributable to Earth's orbit, as shown in Fig. 3(b and c), where seasonality was amplified in the NH and diminished in the Southern Hemisphere."

Page 21, table 2: This table does not provide more information than the description in the text, either remove this table or provide more specific information than only given the reference.

We intended to list up all the experiments with a set of Table 1 and Table 2.
You pointed out that it would be better to have a table of all the experiments to be able to see all the experiments at a glance, so we changed Table 1 to a list of all the experiments by adding LM and HIST.

Line 680, Fig6b: there is a sharp gradient at around 30N, can you explain?

The sharp gradient shown by the contour lines around 32°N would be caused by a strong and deep westerly boundary current and associated strong upwelling (Brady et al., 2013), which can be seen in the previous LGM modelling studies having strong AMOC (Brady et al. 2013, Muglia and Schmittner 2015, Sherriff-Tadano et al. 2018). This is added in the text.

Line 221-225, regarding the HIST part in Fig13, more information about the three ensembles during HIST period are needed. The HIST part in Fig13 is hard to observe and compare. It would be more informative to show another figure only for HIST part, in order to draw the conclusion that the initial conditions for HIST from the end of LM experiment is similar to that from the long PI run, and discuss if this is the case for other models or it might be model dependent.

In Fig. 13(b), we included a figure from 1850 to 2014; Fig. 13(b) includes an additional 30 historical experiments for CMIP6, which are increased ensemble members recently using

MIROC-ES2L.

On the other hand, the historical ensemble experiment was removed from Figure 13(a), making it difficult to identify. The historical ensemble starting from the standard PI had a large positive bias from HadCRUT4 in the late 19th and early 20th century, whereas the post-LM HIST experiment showed a small positive bias. This is shown as a histogram in Figure 13c and is discussed in Section 4.4.

The authors present the four experiments separately, a summary table or figure to compare the four past periods would be helpful to have an overview of the climate change, and differences of modelled glacial and interglacial climate.

We have summarized them in Table 1, as mentioned above.

Minors:
Line 36, "the Pliocene", should be " mid-Pliocene (3.2 million years before present)".

changed

Line 181, "by PI", suggest change to "in PI or at PI".

changed

---

## Author Response (AR2)

**Reply for the editor of PMIP4 experiments using MIROC-ES2L Earth System Model**

Rumi Ohgaito, Akitomo Yamamoto, Tomohiro Hajima, Ryouta O'ishi, Manabu Abe, Hiroaki Tatebe, Ayako Abe-Ouchi, Michio Kawamiya

Editor's comment:
As I understand it (but I am not an expert in this!) the SST data in question are considered to be of very high uncertainty (Hessler et al 2014). Thus you may have unnecessarily weakened your statement here. Whether or not you agree with my appraisal of the SST data, I think it would be better to discuss SAT and SST separately here. For SAT, I think you can basically revert to your original statement.In relation to the SST I will just say that I look forward to your response!

Following the editor's suggestion, we modified the sentences as follows. We also add a citation for a new dataset (Kaufman et al. 2020) that has been published.

> ➢ Compared with PI, temperature over the tropics is generally lower in the 6ka experiment, which may partly be inconsistent with proxy data archives (Bartlein et al., 2011, Kaufman et al., 2020). Because the uncertainty of the marine proxies is so large (Hessler et al., 2014) that it is unlikely to be possible to identify positive or negative SST anomalies at 6ka, we exclude them from our assessment here.

Additionally, the values in table 2 and related numbers in the text have been replaced by more accurate calculations and Figure S3 is replaced with a new figure for improved visibility.

**Marked up manuscript**

[revised manuscript text omitted]

---

## Author Response (AR3)

**Responses to the editor's comments**

**Thank you, editor. Below you will find our reply to each of your points and the manuscript we have changed.**

1. Please either describe how the climate sensitivity of the model is estimated, or add a citation to where this is described.

**Gregory et al. (2004) is added.**

2. What you wrote in the appendix is actually the kind of information that is central to a GMD paper and I think that the narrative would be clearer if the information were included in section 3.2. It is rather disjointed at present. It is also bit unclear exactly what the problem was that had to be overcome. ie it would be better to explain the problem as well as the solution. If you decide to retain an appendix, it needs citing in section 3.2 and shouldn't it be positioned at the end of the manuscript, just before the references?

**Explanations are added in Sect 3.2 (See the following mark-up manuscript) and the Appendix is relocated just before the references. In order to avoid confusion at this last stage by shuffling the figures, we would like to retain the appendix.**

3. Table 1. "Greenhouse" not "Greenhous"

**changed**
* * *

[revised manuscript text omitted]